# The Role of Factor Xa-Independent Pathway and Anticoagulant Therapies in Cancer-Related Stroke

**DOI:** 10.3390/jcm11010123

**Published:** 2021-12-27

**Authors:** Hyung Jun Kim, Jong-Won Chung, Oh Young Bang, Yeon Hee Cho, Yun Jeong Lim, Jaechun Hwang, Woo-Keun Seo, Gyeong-Moon Kim, Hee-Jin Kim, Myung-Ju Ahn

**Affiliations:** 1Department of Neurology, Seoul Hospital, College of Medicine, Ewha Woman’s University, Seoul 07804, Korea; khhhj7@naver.com; 2Department of Neurology, Samsung Medical Center, School of Medicine, Sungkyunkwan University, Seoul 06351, Korea; neurocjw@gmail.com (J.-W.C.); smcyunjeong@naver.com (Y.J.L.); mcastenosis@gmail.com (W.-K.S.); kimgm@skku.edu (G.-M.K.); 3Translational and Stem Cell Research Laboratory on Stroke, Samsung Medical Center, Seoul 06351, Korea; cyeon2@empas.com; 4Department of Neurology, Kyungpook National University Chilgok Hospital, School of Medicine, Kyungpook National University, Daegu 41404, Korea; ghkdwocns@gmail.com; 5Department of Laboratory Medicine and Genetics, Samsung Medical Center, School of Medicine, Sungkyunkwan University, Seoul 06351, Korea; hee_jin.kim@samsung.com; 6Department of Hemato-Oncology, Samsung Medical Center, School of Medicine, Sungkyunkwan University, Seoul 06351, Korea; silk.ahn@samsung.com

**Keywords:** stroke, coagulopathy, thrombosis, cancer, anticoagulation

## Abstract

Background: The optimal strategy for stroke prevention in cancer patients is unknown. We compared the underlying mechanisms of coagulopathy and the effects of anticoagulants in patients with active cancer and atrial fibrillation (AF). Methods: We retrospectively enrolled 46 consecutive patients with embolic stroke of unknown source and active cancer (cancer stroke). We consecutively screened patients with cancer patients without stroke (*n* = 29), AF stroke (*n* = 52), and healthy subjects (*n* = 28), which served as controls. Patients with cancer stroke were treated with either enoxaparin (a low-molecular-weight heparin) or a factor Xa inhibitor, and those with AF stroke were treated with factor Xa inhibitors. D-dimer, factor Xa, and circulating cell-free DNA (cfDNA), a marker of neutrophil extracellular traposis, were measured at both before and after anticoagulation. Results: In AF stroke, factor Xa activity and cfDNA and D-dimer levels were decreased by treatment with factor Xa inhibitors. In contrast, in cancer stroke, factor Xa activity was decreased, D-dimer levels were unchanged, and cfDNA levels were increased by treatment with factor Xa inhibitors. In cancer stroke patients treated with enoxaparin, D-dimer levels were decreased (*p* = 0.011) and cfDNA levels were unchanged. Conclusion: The anticoagulation effects of factor Xa inhibitors differed between cancer stroke and AF stroke.

## 1. Introduction

Systemic cancer and ischemic stroke are common conditions and two of the most frequent causes of death among the elderly. Cancer-related stroke (cancer stroke) is relatively common and should be suspected in patients with embolic stroke of unknown source (ESUS) and active cancer [1,2]. The molecular mechanisms of coagulopathy in cancer stroke remain unclear, although an understanding of these would assist the development of therapeutic strategies against coagulopathy in these patients. A number of prothrombotic mechanisms have been proposed to explain cancer-related coagulopathy, which include the initiation of the extrinsic pathway by clotting initiator protein (tissue factor) and polyphosphate (polyP) in circulating tumor-derived extracellular vesicles (EVs), and the interaction of tumor-derived EVs with platelets or neutrophil extracellular traposis (NETosis) [3]. We have shown that increased circulating cell-free DNA (cfDNA) levels are associated with cancer stroke, suggesting that NETosis is one of the molecular mechanisms of cancer stroke [4].

The effective correction of hypercoagulability as measured by D-dimer levels promotes the survival of cancer stroke patients [5]. Low-molecular-weight heparins (LMWHs) are widely used for the prevention of stroke/systemic embolism (SE) in cancer stroke. However, LMWHs require injection and may not be suitable for long-term maintenance therapy to prevent the recurrence of stroke. Patient persistence with anticoagulants to prevent venous thromboembolism (VTE) is lower in patients receiving LMWHs than in those receiving vitamin K antagonist (VKA) or direct oral anticoagulants (DOACs) [6]. In a recent trial comparing LMWH and aspirin in patients with active cancer and acute ischemic stroke, 60% of patients randomized to LMWH crossed over to aspirin due to discomfort [7]. DOACs could be an alternative option for the treatment of cancer stroke. DOACs are now recommended as the preferred alternative to VKA for reducing the risk of stroke/SE associated with non-valvular atrial fibrillation (AF) [8,9]. More recently, DOACs have also been shown to be as effective as conventional anticoagulant therapy for the treatment of cancer-related VTE [10]. Although rivaroxaban and edoxaban showed an increased bleeding risk especially in gastrointestinal cancer [11], recent randomized controlled trials comparing the effects of LMWHs and DOACs showed that DOACs could be an alternative option for the treatment of cancer patients with VTE [12,13]. Recently published guidelines recommend DOACs as a maintenance therapy in selected patients to prevent VTE [14,15,16].

However, the efficacy and safety of DOACs in cancer stroke have not yet been documented by comprehensive studies. In terms of pathomechanisms, thrombi formed within the vessel in the absence of a gross nidus for thrombus formation/propagation (high flow system), such as intravascular coagulopathy in cancer patients, may differ from thrombi formed in the left atrial appendage in AF patients or a deep vein in cancer patients (low flow system) [17,18]. We hypothesize that the pathomechanisms of coagulopathy may differ between patients with cancer stroke and those with AF stroke. Thus, strategies against cancer stroke may differ depending on the pathomechanisms. To this end, we measure changes in biomarkers for factor Xa-dependent (factor Xa activity) and independent pathways (cfDNAs, a marker of NETosis), and compared the effects of DOACs and LMWHs on D-dimer levels, a marker of coagulopathy.

## 2. Methods

### 2.1. Patients

We retrospectively studied consecutive patients of the OASIS-Cancer study (ClinicalTrials.gov identifier: NCT02743052). The inclusion criteria were (1) acute ischemic embolic stroke documented based on diffusion-weighted imaging (DWI) within 7 days after symptom onset due to ESUS; (2) known or newly diagnosed active cancer at the time of stroke diagnosis, after stroke, or during hospitalization; (3) patients who collect blood samples before anticoagulation administration; (4) hypercoagulability state, defined as D-dimer concentration >3 μg/mL [19]; and (5) no use of warfarin, DOACs, or LMWH prior to stroke. Patients were considered to have active cancer if they were diagnosed with cancer, underwent treatment within 6 months before enrollment, or had recurrent/metastatic cancer. We excluded patients who had conventional stroke mechanisms, such as AF or carotid stenosis, and patients who did not provide blood samples for D-dimer and cfDNA levels, and factor Xa activity before anticoagulation treatment.

Patients admitted for ischemic stroke due to AF at a University Medical Center between April 2015 and August 2020 were recruited consecutively as the AF stroke group. The inclusion criteria were (1) acute ischemic embolic stroke documented on DWI within 7 days after symptom onset, (2) no history of cancer, (3) no use of warfarin or DOACs prior to stroke, and (4) prescription of factor Xa inhibitors after enrollment.

Cancer patients without stroke (cancer controls) and healthy subjects (normal controls) served as control groups. Patients with locally advanced or systemic metastatic lung cancer (mostly adenocarcinoma) without a history of stroke were recruited as the cancer control group. Healthy subjects without a history of stroke or cancer served as the normal control group. All participants or their next of kin provided informed consent for participation in the study. The local Institutional Review Board approved this study (IRB No. SMC 2016-02-104) (Figure 1). 

### 2.2. Work-Up for Stroke

Age, sex, and stroke risk factors, including hypertension, diabetes mellitus, hyperlipidemia, and smoking habits, were obtained for all patients. The type of primary cancer lesion, histology, and presence of systemic metastasis were also recorded. Routine laboratory data were collected for all patients. Both cancer stroke and AF stroke patients underwent electrocardiography, 24 h Holter monitoring or 72 h in-patient telemonitoring, echocardiography, brain magnetic resonance (MR) imaging, and MR angiography. The patterns of acute stroke on DWI were reviewed and classified as single/multiple lesions involving one vascular territory or multiple lesions involving multiple vascular territories by two independent readers (H.J.K. and J.W.C.). Stroke mechanisms were determined using the Causative Classification System [20] at a regular consensus meeting. To find additional thrombotic disease, such as pulmonary thromboembolism and deep vein thrombosis, we did not routinely perform pulmonary computed tomography angiogram or duplex sonography.

### 2.3. Hypercoagulability Measurements

Plasma samples were prepared from citrated whole blood following immediate centrifugation for 15 min at 2000× *g* and stored at −80 °C until further analyses. In this study, cfDNA was evaluated as an NET-specific biomarker, and analyzed using the Quant-iT PicoGreen dsDNA assay (Invitrogen in Carlsbad, California and the Cell Death Detection ELISA Kit (Roche, Rotkreuz, Switzerland). The endogenous factor Xa assay was performed by measuring the resultant factor Xa activity using the STA^®^-Deficient X assay on the STA R MAX^®^ analyzer (Stago, Asnières-sur-Seine, France). The assay consists of measuring the clotting time, in the presence of a Stago PT reagent, of a system in which all the factors are present and in excess (supplied by STA^®^-Deficient X) except factor X, which is derived from the sample being tested. Lastly, the D-dimer level, a marker of hypercoagulability, was measured as per the manufacturer’s protocol. In this study, D-dimer and cfDNA levels, and factor Xa activity were measured at both before anticoagulation and within 30 days after anticoagulation, and at the time of stroke/SE recurrence during follow-up.

The antithrombotic treatment was determined based on the stroke mechanism. In patients with cancer stroke, patients with conventional mechanisms were excluded and hypercoagulable state was confirmed. Therefore, anticoagulation treatment was considered as the first-line antithrombotic treatment in the absence of evident contraindications. Patients received either 1 mg/kg enoxaparin sodium subcutaneously, twice a day, or 60 mg edoxaban orally, once a day. The dose of edoxaban was reduced to 30 mg/day under any of the following conditions: creatinine clearance, 30–50 mL/min; body weight, 60 kg or lower; or concomitant therapy with a strong P-glycoprotein inhibitor. For patients with AF stroke, on-label dosing of factor Xa inhibitors was prescribed. The choice of anticoagulation largely depended on the physician or patient preference. 

### 2.4. Statistical Analysis

Differences in discrete variables among the groups were evaluated using the Chi-squared test, Fisher’s exact test, or the Mann–Whitney test. Differences in continuous variables were analyzed using t-tests or Kruskal–Wallis tests. D-dimer and cfDNA levels, and factor Xa activity were compared among the groups. A paired t-test and Wilcoxon signed rank test was used to compare changes in hypercoagulability measurements before and after anticoagulation treatment. In all analyses, statistical significance was set at *p* < 0.05. All statistical analyses were performed using SPSS (version 23.0; IBM, Chicago, IL, USA) and the open-source statistical package R (version 3.6.3; R Project for Statistical Computing, Vienna, Austria).

## 3. Results

A total of 155 patients were enrolled in this study, excluding 33 patients in whom anticoagulation could not be started/maintained or hypercoagulability measurement could not be measured prior to anticoagulation: 46 patients with cancer stroke, 29 cancer controls, 52 patients with AF stroke, and 28 healthy controls (Figure 1). Of the patients with cancer stroke, 33 (71.7%) received enoxaparin and 13 (28.3%) received edoxaban. Of the patients with AF stroke, 26 (50.0%) received edoxaban, 20 (38.5%) received apixaban, and 6 (11.5%) received rivaroxaban. The mean patient age was 66.3 ± 9.6 years (range, 35–89 years), and 82 were men and 73 were women. The AF stroke group was older than the cancer stroke group (*p* < 0.001), and the cancer stroke group included more male patients than the AF stroke and normal control groups (*p* < 0.001). The risk factors for stroke, such as hypertension and hyperlipidemia, were more prevalent in the AF stroke group than in the cancer stroke group (*p* < 0.001); however, diabetes mellitus and smoking showed the opposite results (*p* < 0.001). The primary cancers in the cancer stroke group included lung (39.1%), hepatobiliary (28.3%), gastrointestinal (13.0%), breast-gynecologic (13.0%), and other cancers (6.5%), while all cancer controls had lung cancer. Adenocarcinoma (*p* = 0.231) and systemic metastasis (*p* = 0.357) were not different between two cancer groups (cancer stroke vs. cancer control) (Table 1). Patient characteristics according to anticoagulation (enoxaparin vs. edoxaban) are summarized in Data II in the Data Supplement.

### 3.1. Baseline D-Dimer and cfDNA Levels, and Factor Xa Activity among the Groups

D-dimer and cfDNA levels were elevated in stroke patients (both the cancer stroke and AF stroke groups) compared with the non-stroke control groups, while factor Xa activity was reduced in the former compared with the latter (*p* < 0.05 for all cases). D-dimer and cfDNA levels were higher in patients with cancer stroke than in those with AF stroke (*p* < 0.05 for both). Factor Xa activity was lower in patients with cancer stroke than in those with AF stroke (*p* = 0.041). Compared to the AF stroke and cancer control groups, patients in the cancer stroke group had lower platelet counts, and prolonged prothrombin time/international normalized ratio (PT/INR) and activated partial thromboplastin time (aPTT) (*p* < 0.001 for all cases) (Table 1).

### 3.2. D-Dimer and cfDNA Levels, and Factor Xa Activity in Recurrent Cancer Stroke

Stroke recurrence was observed in 10 patients with active cancer and acute stroke; the mean time from enrollment to recurrence was 20.4 ± 30.7 days (range, 3–105 days). When D-dimer and cfDNA levels, and factor Xa activity were compared prior to (stable state) and at the time of stroke recurrence, the recurrence of cancer stroke was associated with elevation of D-dimer and cfDNA levels (*p* < 0.05 for both cases), but not with factor Xa activity levels (Appendix A, Figure 2).

### 3.3. Changes in D-Dimer and cfDNA Levels, and Factor Xa Activity after Anticoagulation

The time intervals from baseline to follow-up measurement were not significantly different between those treated with enoxaparin and those treated with edoxaban (9.3 ± 12.6 days and 6.8 ± 2.5 days, respectively, *p* = 0.301). Baseline levels of D-dimer, cfDNA, and factor Xa activity were comparable between cancer stroke patients treated with enoxaparin and edoxaban. However, changes in D-dimer and cfDNA levels were different depending on the types of anticoagulants received by patients with cancer stroke. cfDNA levels were significantly increased and D-dimer levels were not significantly reduced after treatment with edoxaban, while D-dimer levels were dramatically decreased by approximately half (*p* = 0.011) and cfDNA levels were unchanged after treatment with enoxaparin.

In contrast, D-dimer and cfDNA levels were significantly decreased after treatment with factor Xa inhibitors in patients in the AF stroke group. Factor Xa activity was significantly decreased by treatment with factor Xa inhibitors in both the AF stroke and cancer stroke groups (Appendix A, Figure 2).

## 4. Discussion

This study shows differences in the coagulation profile and response to factor Xa inhibitors between cancer stroke and AF stroke. D-dimer and cfDNA levels were higher, while factor Xa activity was lower in cancer stroke than in AF stroke. In both cancer stroke and AF stroke, the use of factor Xa inhibitors reduced factor Xa activity. However, unlike in AF stroke, D-dimer levels were unchanged and cfDNA levels were increased by treatment with factor Xa inhibitors in cancer stroke.

In AF stroke, coagulation factor Xa plays an important role, and target inhibitors have been developed to prevent the recurrence of stroke/SE [21]. The present study is the first to show that factor Xa inhibitors reduce factor Xa activity and D-dimer levels in patients with AF stroke. However, in cancer stroke, the elevated D-dimer levels were not significantly decreased by the use of a factor Xa inhibitor, even though factor Xa activity was decreased significantly. Therefore, unlike AF stroke, the factor Xa-independent coagulation pathway might play a pivotal role in coagulopathies in cancer stroke. 

In addition to factor Xa-dependent coagulation pathways triggered by tissue factors, circulating cancer cell-derived EVs and NETosis have recently emerged as molecular mechanisms of coagulopathy in cancer stroke [22,23,24]. In a previous study, cfDNAs, a NETosis marker, were independently associated with cancer stroke and D-dimer levels [4]. In the present study, cfDNA levels were elevated in patients with cancer stroke compared with AF stroke and elevated cfDNA levels were associated with recurrent cancer stroke. With the use of factor Xa inhibitors, cfDNA levels were increased in cancer stroke, but decreased in AF stroke. With the use of an LMWH, D-dimer levels were dramatically decreased, and cfDNA levels and factor Xa remained stable in cancer stroke. These results emphasize the importance of controlling NETosis over factor Xa activity in cancer stroke. Recently, there have been various studies aimed at suppressing NETosis in various pathological situations, such as the study of inhibitor of peptidylarginine deiminase 4, an enzyme highly expressed in cancer and neutrophils [25,26]. In addition, in previous studies, LMWH was shown to have anti-inflammatory activity unlike factor Xa inhibitors [27]. Because the reasons for increased stroke risk in cancer patients are multifactorial, further studies are needed that evaluate blood tests for other molecular mechanisms of coagulopathy, besides factor Xa activity, and D-dimer and cfDNA levels.

This study has some limitations. First, the sample size in this study was relatively small. As a result, many factors, such as cancer treatment, were not considered in detail. Chemotherapy induces cell death and DNA release into the plasma, but the duration, type, and regimen of chemotherapy was not considered in our analysis. A larger prospective study is needed to confirm our findings. Second, in our study, there were differences in primary lesions between the cancer stroke and the cancer control group and that may influence the results of our study. However, there was no difference in histopathology and systemic involvement between the two groups. Third, we found that factor Xa activity in stroke patients (cancer stroke and AF stroke) was lower than that in patients without stroke (cancer controls and normal controls), which was an unexpected finding. To our knowledge, there have been no studies evaluating factor Xa activity in stroke patients. Therefore, it is difficult to accurately explain this finding, but it may be related to a longer PT/INR and aPTT and a lower platelet count in cancer stroke compared with cancer controls [28]. Fourth, although this study enrolled consecutive patients, factor Xa activity and cfDNA levels were not measured in some of the patients due to medical and technical reasons (trial and error in measurements during the early study period), possibly leading to selection bias. Fifth, in this study, the role of antiplatelets in factor Xa-dependent and independent pathways was not investigated. Many centers treat patients with cancer-related stroke with antiplatelet therapy. In addition, a recent subgroup analysis of the NAVIGATE-ESUS trial found that anticoagulation with rivaroxaban was not superior and perhaps even inferior to antiplatelet therapy with aspirin in patients with history of cancer and ESUS [29]. Further studies are needed. Sixth, long-term clinical outcomes were not measured in this study because patients with cancer stroke showed a high mortality rate. Additionally, in some patients with cancer stroke the kind of anticoagulant was changed due to elevation of D-dimer levels or the occurrence of embolic events/thrombocytopenia. Therefore, only short-term laboratory changes could be confirmed before the anticoagulant was changed or discontinued. Seventh, in this study, the level of prothrombin fragment 1.2 (F1.2) was not measured. It deserves further studies on the changes of prothrombin fragment 1.2 (F1.2) with the use of DOACs and LMWH in cancer stroke. Last but not the least, the results of this study should be interpreted with caution, because cancer stroke can be caused by clots originating from the venous system (paradoxical embolism from DVT), cardiac valves (non-bacterial thrombotic endocarditis), or arteries (intravascular coagulopathy) [1]. Therefore, coagulation abnormalities in cancer patients may differ depending on the source of clots and further studies are needed.

## 5. Conclusions

This study measured changes in biomarkers for factor Xa-dependent and independent pathways and showed differential anticoagulation effects of factor Xa inhibitors between cancer stroke and AF stroke. Further studies are needed to determine whether factor Xa inhibitors are non-inferior to LMWH in the prevention of recurrent stroke/SE in patients with cancer-related stroke. Two prospective randomized controlled trials are ongoing, including the ENCHASE study (Edoxaban for the Treatment of Coagulopathy in Patients with Active Cancer and Acute Ischemic Stroke, ClinicalTrial.gov, identifier NCT03570281), comparing edoxaban and LMWH in cancer-related stroke, and TEACH2 (Trial of Apixaban versus Aspirin in Cancer Patients with Cryptogenic Ischemic Stroke), comparing apixaban with aspirin in preventing major thromboembolic events and venous thromboembolism.

## Figures and Tables

**Figure 1 jcm-11-00123-f001:**
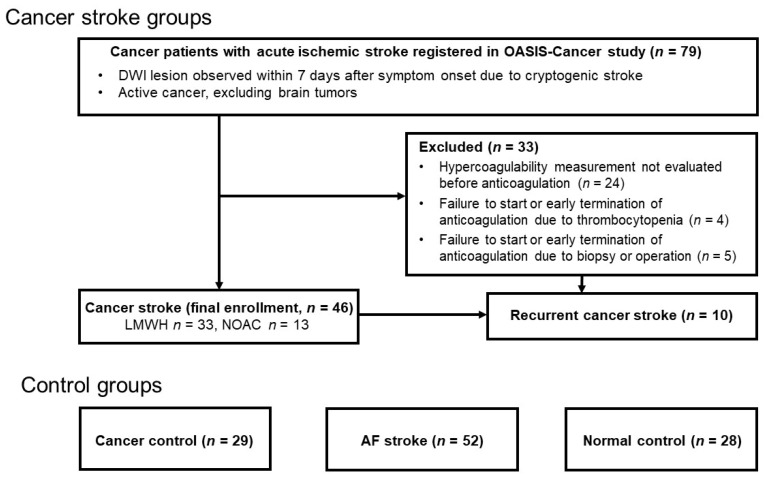
Flow chart. DWI, diffusion-weighted image; AF atrial fibrillation; LMWH, low-molecular-weight heparin; DOAC, direct oral anticoagulant.

**Figure 2 jcm-11-00123-f002:**
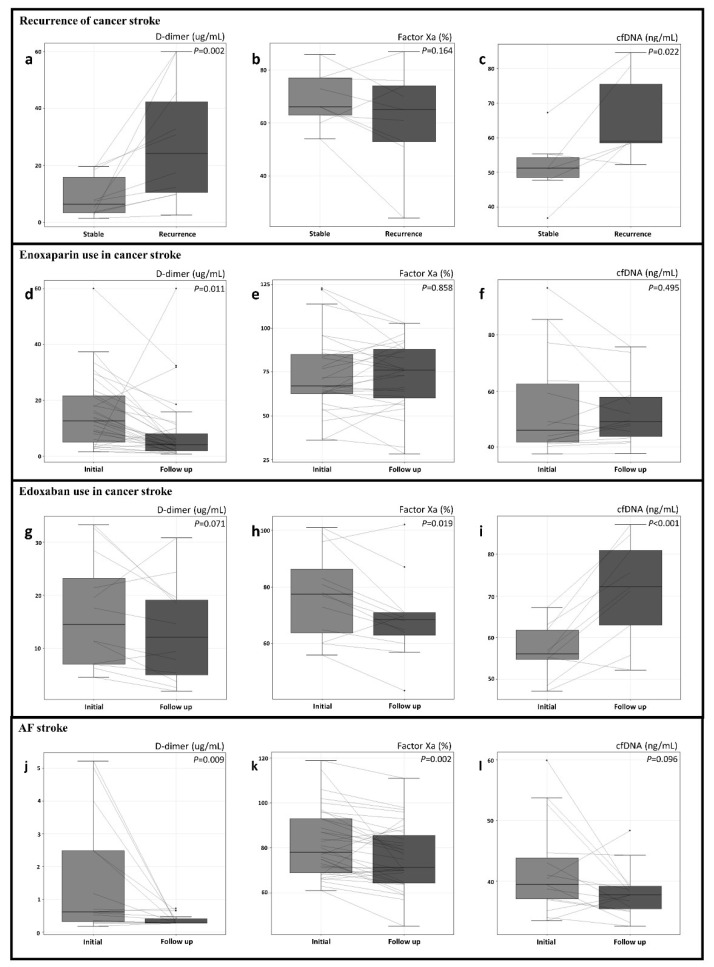
Changes in D-dimer, factor Xa, and circulating cell-free DNA (cfDNA) levels after recurrence of stroke in patients with cancer stroke (**a**–**c**) and the use of enoxaparin (**d**–**f**) or edoxaban (**g**–**i**) in patients with cancer stroke and atrial fibrillation stroke (**j**–**l**).

**Table 1 jcm-11-00123-t001:** Baseline characteristics of the subjects.

				Groups					*P*	
	Cancer Stroke (*CS*)			Cancer Control (*CC*, *N* = 29)	AF Stroke(*AF*, *N* = 52)	Normal Control (*NC*, *N* = 28)	*LMWH vs.* *DOAC*	*CS vs.* *CC*	*CS vs.* *AF*	*CS vs.* *NC*
	Total (*N* = 46)	*LMWH* (*N* = 33)	*DOAC* (*N* = 13)							
Age (years)	64.4 ± 10.3	64.5 ± 10.8	64.0 ± 9.1	64.1 ± 9.6	71.5 ± 8.8	61.8 ± 4.4	0.593	0.910	<0.001	0.144
Male	32 (69.6%)	22 (66.7%)	10 (76.9%)	15 (51.7%)	28 (53.9%)	7 (25.0%)	0.817	0.190	<0.001	<0.001
Risk factor										
Hypertension	19 (41.3%)	15 (45.5%)	4 (30.8%)	7 (24.1%)	30 (57.7%)	5 (17.9%)	0.641	0.203	<0.001	0.067
Diabetes	10 (21.7%)	5 (15.2%)	5 (38.5%)	1 (3.5%)	8 (15.4%)	1 (3.6%)	0.152	0.065	<0.001	0.073
Hyperlipidemia	9 (19.6%)	7 (21.2%)	2 (15.4%)	1 (3.5%)	22 (42.3%)	1 (3.6%)	>0.99	0.099	<0.001	0.109
Current smoking	11 (23.9%)	9 (27.3%)	2 (15.4%)	8 (27.6%)	4 (7.7%)	2 (7.1%)	0.696	0.933	<0.001	0.128
Cancer type										
Adenocarcinoma	31 (67.4%)	22 (66.7%)	9 (69.2%)	24 (82.8%)	*N/A*	*N/A*	>0.99	0.231	*N/A*	*N/A*
Systemic metastasis	33 (71.7%)	23 (69.7%)	10 (76.9%)	17 (58.6%)	*N/A*	*N/A*	0.973	0.357	*N/A*	*N/A*
Primary cancer lesion							0.638	*N/A*	*N/A*	*N/A*
Lung	18 (39.1%)	11 (33.3%)	7 (53.9%)	29 (100.0%)	*N/A*	*N/A*				
Gastrointestinal	6 (13.0%)	5 (15.2%)	1 (7.7%)	0 (0.0%)	*N/A*	*N/A*				
Hepatobiliary	13 (28.3%)	10 (30.3%)	3 (23.1%)	0 (0.0%)	*N/A*	*N/A*				
Breast-gynecology	6 (13.0%)	4 (12.1%)	2 (15.4%)	0 (0.0%)	*N/A*	*N/A*				
Other	3 (6.5%)	3 (9.1%)	0 (0.0%)	0 (0.0%)	*N/A*	*N/A*				
Treatment within 4 weeks before enrollment										
Surgery	5 (10.9%)	4 (12.1%)	1 (7.7%)	3 (10.3%)	*N/A*	*N/A*	0.999	0.999	*N/A*	*N/A*
Chemotherapy	24 (52.2%)	18 (54.6%)	6 (46.2%)	13 (44.8%)	*N/A*	*N/A*	0.853	0.702	*N/A*	*N/A*
Radiotherapy	10 (21.7%)	6 (18.2%)	4 (30.8%)	5 (17.2%)	*N/A*	*N/A*	0.593	0.859	*N/A*	*N/A*
Laboratory results										
D-dimer, μg/mL	19.0 ± 15.1 (*n* = 46)	16.8 ± 16.6 (*n* = 33)	17.0 ± 11.0 (*n* = 13)	0.8 ± 1.3 (*n* = 23)	1.5 ± 1.9 (*n* = 39)	0.4 ± 0.3 (*n* = 28)	0.863	<0.001	<0.001	<0.001
Factor Xa, %	70.8 ± 23.4 (*n* = 44)	69.4 ± 25.0 (*n* = 31)	74.2 ± 19.3 (*n* = 13)	95.5 ± 12.6 (*n* = 29)	79.2 ± 14.2 (*n* = 52)	98.1 ± 13.5 (*n* = 28)	0.245	<0.001	0.041	<0.001
cfDNA, ng/mL	56.6 ± 16.7 (*n* = 29)	55.7 ± 19.4 (*n* = 19)	58.3 ± 10.6 (*n* = 10)	37.2 ± 5.0 (*n* = 17)	43.2 ± 7.9 (*n* = 43)	38.6 ± 6.6 (*n* = 28)	0.633	<0.001	<0.001	<0.001
Platelet count, 10^3^/μL	165.2 ± 89.5	172.2 ± 92.9	147.6 ± 81.1	222.0 ± 82.1	215.2 ± 70.1	*N/A*	0.301	<0.001	<0.001	*N/A*
	(*n* = 46)	(*n* = 33)	(*n* = 13)	(*n* = 29)	(*n* = 52)					
PT (INR)	1.4 ± 0.6	1.4 ± 0.7	1.3 ± 0.4	1.0 ± 0.1	1.0 ± 0.1	*N/A*	0.301	<0.001	<0.001	*N/A*
	(*n* = 46)	(*n* = 33)	(*n* = 13)	(*n* = 29)	(*n* = 52)					
aPTT (sec)	43.7 ± 26.5	46.1 ± 30.8	37.6 ± 6.8	34.1 ± 3.4	34.4 ± 3.7	*N/A*	0.151	<0.001	<0.001	*N/A*
	(*n* = 46)	(*n* = 33)	(*n* = 13)	(*n* = 29)	(*n* = 52)					

LMWH, low-molecular-weight heparin; DOAC, direct oral anticoagulant; AF, atrial fibrillation; GI, Gastrointestinal; cfDNA, circulating cell-free DNA; PT, prothrombin time; INR, international normalized ratio; aPTT, activated partial thromboplastin time.

## Data Availability

The data presented in this study are available on request from the corresponding authors. The data are not publicly available due to privacy.

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
