# Peer review of "The Role of Factor Xa-Independent Pathway and Anticoagulant Therapies in Cancer-Related Stroke"

_jcm, 2021, doi:10.3390/jcm11010123_

Round 1

Reviewer 1 Report

Major comments:

The study is interesting but it is difficult to draw any conclusions due to the way the statistics are carried out. There are many study groups with completely different baseline characteristics and receive different treatments. At the statistical level, it would be necessary to carry out a model with all these variables, which may not be possible given the sample size. This is important as, for example, rivaroxaban has been shown to have a cardioprotective effect in hypertensive patients. There is also evidence that exposure of platelets to high cholesterol concentrations resulted in loss of responsiveness to factor Xa. This differences in basal characteristics among the groups may lead to wrong conclusions. If statistics couldn´t be repeated in a more accurate way, the authors should look for a comparable cohort of patients in terms of risk factors and/or treatment.

Minor coments:

Abstract should be revised. Background is missing and they only specify the aim of the study.

Tables should also clarify the groups according to different treatments.

Author Response

Response to Reviewers’ comments

December 18, 2021

Manuscript ID: jcm-1497215

Title: Role of Factor Xa-Independent Pathway and Anticoagulant therapies in Cancer-related Stroke

Dear reviewers & editor

First of all, we would like to thank the reviewers and editorial board members for their time and valuable comments on our manuscript. We have addressed our opinions on each comment from two reviewers in this response letter, and made several changes and corrections to our original manuscript.

We tried to revise our manuscript according to reviewer’s suggestions as much as possible, and revised parts are written in red color texts in the manuscript. We hope that the revisions in the manuscript and our accompanying responses will be sufficient to make our manuscript suitable for publication in Journal of Clinical Medicine.

We shall look forward to hearing from you at your earliest convenience.

Sincerely yours,

Oh Young Bang, MD, PhD

Department of Neurology, Samsung Medical Center, Sungkyunkwan University,

50 Irwon-dong, Gangnam-gu, Seoul 135-710, Korea

Tel.: 82-2-3410-3599; Fax: 82-2-3410-1430;

Reviewer 1

The study is interesting but it is difficult to draw any conclusions due to the way the statistics are carried out.

Comment 1. There are many study groups with completely different baseline characteristics and receive different treatments. At the statistical level, it would be necessary to carry out a model with all these variables, which may not be possible given the sample size. This is important as, for example, rivaroxaban has been shown to have a cardioprotective effect in hypertensive patients. There is also evidence that exposure of platelets to high cholesterol concentrations resulted in loss of responsiveness to factor Xa. This differences in basal characteristics among the groups may lead to wrong conclusions. If statistics couldn´t be repeated in a more accurate way, the authors should look for a comparable cohort of patients in terms of risk factors and/or treatment.

Answer 1. We fully agree with the reviewer’s concern. In our study, multivariable analysis was not performed due to the limitation of the number of samples. However, there was no significant difference in basal characteristics compared with cancer control, which may compensate for the limitation. In addition, we classified cancer stroke patients into LMWH and DOAC, and newly described in Table 1 in detail. There was no difference in risk factors or cancer treatment between the two groups. 

Table 1. Baseline characteristics of the subjects.

Groups

P

Cancer stroke (CS)

Cancer control
(CC, N=29)

AF stroke
(AF, N=52)

Normal control
(NC, N=28)

LMWH vs.

DOAC

CS vs.

CC

CS vs.

AF

CS vs.

NC

Total

(N=46)

LMWH

(N=33)

DOAC

(N=13)

Age (years)

64.4±10.3

64.5±10.8

64.0±9.1

64.1±9.6

71.5±8.8

61.8±4.4

0.593

0.910

<0.001

0.144

Male

32 (69.6%)

22 (66.7%)

10 (76.9%)

15 (51.7%)

28 (53.9%)

7 (25.0%)

0.817

0.190

<0.001

<0.001

Risk factor

Hypertension

19 (41.3%)

15 (45.5%)

4 (30.8%)

7 (24.1%)

30 (57.7%)

5 (17.9%)

0.641

0.203

<0.001

0.067

Diabetes

10 (21.7%)

5 (15.2%)

5 (38.5%)

1 (3.5%)

8 (15.4%)

1 (3.6%)

0.152

0.065

<0.001

0.073

Hyperlipidemia

9 (19.6%)

7 (21.2%)

2 (15.4%)

1 (3.5%)

22 (42.3%)

1 (3.6%)

>0.99

0.099

<0.001

0.109

Current smoking

11 (23.9%)

9 (27.3%)

2 (15.4%)

8 (27.6%)

4 (7.7%)

2 (7.1%)

0.696

0.933

<0.001

0.128

Cancer type

Adenocarcinoma

31 (67.4%)

22 (66.7%)

9 (69.2%)

24 (82.8%)

N/A

N/A

>0.99

0.231

N/A

N/A

Systemic metastasis

33 (71.7%)

23 (69.7%)

10 (76.9%)

17 (58.6%)

N/A

N/A

0.973

0.357

N/A

N/A

Primary cancer lesion

0.638

N/A

N/A

N/A

Lung

18 (39.1%)

11 (33.3%)

7 (53.9%)

29 (100.0%)

N/A

N/A

Gastrointestinal

6 (13.0%)

5 (15.2%)

1 (7.7%)

0 (0.0%)

N/A

N/A

Hepatobiliary

13 (28.3%)

10 (30.3%)

3 (23.1%)

0 (0.0%)

N/A

N/A

Breast-gynecology

6 (13.0%)

4 (12.1%)

2 (15.4%)

0 (0.0%)

N/A

N/A

Other

3 (6.5%)

3 (9.1%)

0 (0.0%)

0 (0.0%)

N/A

N/A

Treatment within 4 weeks before enrollment

Surgery

5 (10.9%)

4 (12.1%)

1 (7.7%)

3 (10.3%)

N/A

N/A

0.999

0.999

N/A

N/A

Chemotherapy

24 (52.2%)

18 (54.6%)

6 (46.2%)

13 (44.8%)

N/A

N/A

0.853

0.702

N/A

N/A

Radiotherapy

10 (21.7%)

6 (18.2%)

4 (30.8%)

5 (17.2%)

N/A

N/A

0.593

0.859

N/A

N/A

Laboratory results

D-dimer, μg/mL

19.0±15.1 (n=46)

16.8±16.6 (n=33)

17.0±11.0 (n=13)

0.8±1.3 (n=23)

1.5 ± 1.9 (n=39)

0.4±0.3 (n=28)

0.863

<0.001

<0.001

<0.001

Factor Xa, %

70.8±23.4 (n=44)

69.4±25.0 (n=31)

74.2±19.3 (n=13)

95.5±12.6 (n=29)

79.2±14.2 (n=52)

98.1±13.5 (n=28)

0.245

<0.001

0.041

<0.001

cfDNA, ng/mL

56.6±16.7 (n=29)

55.7±19.4 (n=19)

58.3±10.6 (n=10)

37.2±5.0 (n=17)

43.2±7.9 (n=43)

38.6±6.6 (n=28)

0.633

<0.001

<0.001

<0.001

Platelet count, 103/μL

165.2±89.5

172.2±92.9

147.6±81.1

222.0±82.1

215.2±70.1

N/A

0.301

<0.001

<0.001

N/A

(n=46)

(n=33)

(n=13)

(n=29)

(n=52)

PT (INR)

1.4±0.6

1.4±0.7

1.3±0.4

1.0±0.1

1.0±0.1

N/A

0.301

<0.001

<0.001

N/A

(n=46)

(n=33)

(n=13)

(n=29)

(n=52)

aPTT (sec)

43.7±26.5

46.1±30.8

37.6±6.8

34.1±3.4

34.4±3.7

N/A

0.151

<0.001

<0.001

N/A

(n=46)

(n=33)

(n=13)

(n=29)

(n=52)

LMWH, low molecular weighted heparin; DOAC, direct oral anticoagulant; AF, atrial fibrillation; GI, Gastrointestinal; cfDNA, circulating cell-free DNA; PT, prothrombin time; INR, international normalized ratio; aPTT, activated partial thromboplastin time.

Reviewer 2 Report

This is a very interesting effort to look at the pathophysiology of stroke in cancer. The introduction provides a good overview of the state of understanding of stroke risk and coagulopathy in cancer and leads naturally into the rationale for the study.

The methods are appropriate but some clarification would be helpful. The definition of active cancer includes treatment within 6 months, which could be irrelevant for disease treated with minimal or no adjuvant therapy. This is addressed in part in the limitations, which acknowledge that chemotherapy regimens were not considered. In 2.3, the treatment options are presented, but it should be explicitly stated that it was at the discretion of the investigator (ie. non-random assignment between Enoxaparin and Edoxaban).

Figure 1 would be more appropriately included in the results section with a brief narrative description of the distribution of patients are reasons for exclusion. The statement and statistical significance of all cancer controls having lung cancer is irrelevant, since this is by design from the inclusion criteria.

In 3.1, there would be value to an accompanying table to illustrate the biomarkers between the different groups. In 3.2 and 3.3, the table is not necessary; the figure is a much better illustration of the comparisons being discussed. Figure 3 is not referenced in the text and does not add materially to the discussion other than as a case example of Edoxaban failure; I am not convinced this is really a recurrence, and would suggest omitting it.

The discussion provides a good validation of the findings, and the limitations are acknowledged well. The final conclusion supports the need for further understanding of the mechanism of stroke to choose the most appropriate therapiies.

Author Response

Response to Reviewers’ comments

December 18, 2021

Manuscript ID: jcm-1497215

Title: Role of Factor Xa-Independent Pathway and Anticoagulant therapies in Cancer-related Stroke

Dear reviewers & editor

First of all, we would like to thank the reviewers and editorial board members for their time and valuable comments on our manuscript. We have addressed our opinions on each comment from two reviewers in this response letter, and made several changes and corrections to our original manuscript.

We tried to revise our manuscript according to reviewer’s suggestions as much as possible, and revised parts are written in red color texts in the manuscript. We hope that the revisions in the manuscript and our accompanying responses will be sufficient to make our manuscript suitable for publication in Journal of Clinical Medicine.

We shall look forward to hearing from you at your earliest convenience.

Sincerely yours,

Oh Young Bang, MD, PhD

Department of Neurology, Samsung Medical Center, Sungkyunkwan University,

50 Irwon-dong, Gangnam-gu, Seoul 135-710, Korea

Tel.: 82-2-3410-3599; Fax: 82-2-3410-1430;

Reviewer 2.

This is a very interesting effort to look at the pathophysiology of stroke in cancer. The introduction provides a good overview of the state of understanding of stroke risk and coagulopathy in cancer and leads naturally into the rationale for the study. The methods are appropriate but some clarification would be helpful.

Comment 1. The definition of active cancer includes treatment within 6 months, which could be irrelevant for disease treated with minimal or no adjuvant therapy. This is addressed in part in the limitations, which acknowledge that chemotherapy regimens were not considered.

Answer 1. We appreciate this insightful comment. In this study, we followed the definition of active cancer was defined as a diagnosis of cancer within 6 months prior to enrollment, any treatment for cancer within the previous 6 months, or recurrent or metastatic cancer, as previously described.[1] However, the duration, type, and regimen of chemotherapy were not analyzed. As per your suggestion, we revised our manuscript from,

“Chemotherapy induces cell death and DNA release into the plasma, but the duration of chemotherapy was not considered in our analysis.”

To

“Chemotherapy induces cell death and DNA release into the plasma, but the duration, type, and regimen of chemotherapy was not considered in our analysis.”

At page # 8 in Limitation section.

Comment 2. In 2.3, the treatment options are presented, but it should be explicitly stated that it was at the discretion of the investigator (ie. non-random assignment between Enoxaparin and Edoxaban).

Answer 2. Thank you for this helpful suggestion. As per your suggestion, we added the following sentence to the Methods section.

“The choice of anticoagulation largely depended on the physician or patient preference.”

At page # 4, paragraph # 6 in Methods section.

Comment 3. Figure 1 would be more appropriately included in the results section with a brief narrative description of the distribution of patients are reasons for exclusion.

Answer 3. We appreciate this insightful comment. Following the reviewer’s suggestion, we moved Figure 1 to the Results section and revised our manuscript from 

“A total of 155 patients were enrolled in this study: 46 patients with cancer stroke, 29 cancer controls, 52 patients with AF stroke, and 28 healthy controls”

To

“A total of 155 patients were enrolled in this study, excluding 33 patients in whom anticoagulation could not be started/maintained or hypercoagulability measurement could not be measured prior to anticoagulation: 46 patients with cancer stroke, 29 cancer controls, 52 patients with AF stroke, and 28 healthy controls (Figure 1)”

Comment 4. The statement and statistical significance of all cancer controls having lung cancer is irrelevant, since this is by design from the inclusion criteria.

Answer 4. Thank you for this helpful suggestion. We revised our manuscript from,

“The primary cancers in the cancer stroke group included lung (39.1%), hepatobiliary (28.3%), gastrointestinal (13.0%), breast-gynecologic (13.0%), and other cancers (6.5%), while all cancer controls had lung cancer (p<0.001).”

To

“The primary cancers in the cancer stroke group included lung (39.1%), hepatobiliary (28.3%), gastrointestinal (13.0%), breast-gynecologic (13.0%), and other cancers (6.5%), while all cancer controls had lung cancer.”

At page # 4, paragraph # 1 in Results section.

And we also revised Table 1.

Table 1. Baseline characteristics of the subjects.

Groups

P

Total Cancer stroke
(CS, N=46)

Enoxaparin

use in

cancer stroke (LMWH, N=33)

Edoxaban

use in

cancer stroke (DOAC, N=13)

Cancer control
(CC, N=29)

AF stroke
(AF, N=52)

Normal control
(NC, N=28)

LMWH vs.

DOAC

CS vs.

CC

CS vs.

AF

CS vs.

NC

Age (years)

64.4±10.3

64.5±10.8

64.0±9.1

64.1±9.6

71.5±8.8

61.8±4.4

0.593

0.910

<0.001

0.144

Male

32 (69.6%)

22 (66.7%)

10 (76.9%)

15 (51.7%)

28 (53.9%)

7 (25.0%)

0.817

0.190

<0.001

<0.001

Risk factor

Hypertension

19 (41.3%)

15 (45.5%)

4 (30.8%)

7 (24.1%)

30 (57.7%)

5 (17.9%)

0.641

0.203

<0.001

0.067

Diabetes

10 (21.7%)

5 (15.2%)

5 (38.5%)

1 (3.5%)

8 (15.4%)

1 (3.6%)

0.152

0.065

<0.001

0.073

Hyperlipidemia

9 (19.6%)

7 (21.2%)

2 (15.4%)

1 (3.5%)

22 (42.3%)

1 (3.6%)

>0.99

0.099

<0.001

0.109

Current smoking

11 (23.9%)

9 (27.3%)

2 (15.4%)

8 (27.6%)

4 (7.7%)

2 (7.1%)

0.696

0.933

<0.001

0.128

Cancer type

Adenocarcinoma

31 (67.4%)

22 (66.7%)

9 (69.2%)

24 (82.8%)

N/A

N/A

>0.99

0.231

N/A

N/A

Systemic metastasis

33 (71.7%)

23 (69.7%)

10 (76.9%)

17 (58.6%)

N/A

N/A

0.973

0.357

N/A

N/A

Primary cancer lesion

0.638

N/A

N/A

N/A

Lung

18 (39.1%)

11 (33.3%)

7 (53.9%)

29 (100.0%)

N/A

N/A

Gastrointestinal

6 (13.0%)

5 (15.2%)

1 (7.7%)

0 (0.0%)

N/A

N/A

Hepatobiliary

13 (28.3%)

10 (30.3%)

3 (23.1%)

0 (0.0%)

N/A

N/A

Breast-gynecology

6 (13.0%)

4 (12.1%)

2 (15.4%)

0 (0.0%)

N/A

N/A

Other

3 (6.5%)

3 (9.1%)

0 (0.0%)

0 (0.0%)

N/A

N/A

Treatment within 4 weeks before enrollment

Surgery

5 (10.9%)

4 (12.1%)

1 (7.7%)

3 (10.3%)

N/A

N/A

0.999

0.999

N/A

N/A

Chemotherapy

24 (52.2%)

18 (54.6%)

6 (46.2%)

13 (44.8%)

N/A

N/A

0.853

0.702

N/A

N/A

Radiotherapy

10 (21.7%)

6 (18.2%)

4 (30.8%)

5 (17.2%)

N/A

N/A

0.593

0.859

N/A

N/A

Laboratory results

D-dimer, μg/mL

19.0±15.1 (n=46)

16.8±16.6 (n=33)

17.0±11.0 (n=13)

0.8±1.3 (n=23)

1.5 ± 1.9 (n=39)

0.4±0.3 (n=28)

0.863

<0.001

<0.001

<0.001

Factor Xa, %

70.8±23.4 (n=44)

69.4±25.0 (n=31)

74.2±19.3 (n=13)

95.5±12.6 (n=29)

79.2±14.2 (n=52)

98.1±13.5 (n=28)

0.245

<0.001

0.041

<0.001

cfDNA, ng/mL

56.6±16.7 (n=29)

55.7±19.4 (n=19)

58.3±10.6 (n=10)

37.2±5.0 (n=17)

43.2±7.9 (n=43)

38.6±6.6 (n=28)

0.633

<0.001

<0.001

<0.001

Platelet count, 103/μL

165.2±89.5

172.2±92.9

147.6±81.1

222.0± 82.1

215.2±70.1

N/A

0.301

<0.001

<0.001

N/A

(n=46)

(n=33)

(n=13)

(n=29)

(n=52)

PT (INR)

1.4±0.6

1.4±0.7

1.3±0.4

1.0 ± 0.1

1.0 ± 0.1

N/A

0.301

<0.001

<0.001

N/A

(n=46)

(n=33)

(n=13)

(n=29)

(n=52)

aPTT (sec)

43.7±26.5

46.1±30.8

37.6±6.8

34.1 ± 3.4

34.4 ±3.7

N/A

0.151

<0.001

<0.001

N/A

(n=46)

(n=33)

(n=13)

(n=29)

(n=52)

LMWH, low molecular weighted heparin; DOAC, direct oral anticoagulant; AF, atrial fibrillation; GI, Gastrointestinal; cfDNA, circulating cell-free DNA; PT, prothrombin time; INR, international normalized ratio; aPTT, activated partial thromboplastin time

Comment 5. In 3.1, there would be value to an accompanying table to illustrate the biomarkers between the different groups.

Answer 5. Thank you for this helpful suggestion. The results described in 3.1 are included in Table 1. Therefore, we revised our manuscript from,

“Compared to the AF stroke and cancer control groups, patients in the cancer stroke group had lower platelet counts, and prolonged prothrombin time/international normalized ratio (PT/INR) and activated partial thromboplastin time (aPTT) (p < 0.001 for all cases).”

To

“Compared to the AF stroke and cancer control groups, patients in the cancer stroke group had lower platelet counts, and prolonged prothrombin time/international normalized ratio (PT/INR) and activated partial thromboplastin time (aPTT) (p < 0.001 for all cases) (Table 1).”

At page # 6, paragraph # 2 in Results section.

Comment 6. In 3.2 and 3.3, the table is not necessary; the figure is a much better illustration of the comparisons being discussed.

Answer 6. We appreciate this insightful comment. We fully agree with the reviewer’s opinion that Figure 2 represents the results better than Table 2. Therefore, we will change Table 2 to Supplemental Data 1. In addition, the size of the text font and color in Figure 2 has been revised so that it can be seen better.  

Comment 7. Figure 3 is not referenced in the text and does not add materially to the discussion other than as a case example of Edoxaban failure; I am not convinced this is really a recurrence, and would suggest omitting it.

Answer 7. We appreciate this insightful comment. We completely agree with the reviewer’s opinion. Therefore, we will delete it from the main body and show it only in Supplemental data.

The discussion provides a good validation of the findings, and the limitations are acknowledged well. The final conclusion supports the need for further understanding of the mechanism of stroke to choose the most appropriate therapiies.

Reviewer 3 Report

I have read with interest this study dealing with both cancer and non-cancer stroke. Although the sample size is rather small, results deserve attention. However, I have some important critical points to be addressed to the Authors.

1 Page 2, line 55. Direct Oral Anticoagulants (DOAC) is a better term than NOAC.

2 Page 2, line 60. The authors should be remember here that some DOAC (Edoxaban and Rivatroxaban) confer an increased risk of bleeding in patients with cancer of the GI and Urinary tracts. References related to those studies should be added.

3 Page 4, line 125. Why did the authors choose factor Xa activity as a test for a hypercoagulative condition? It is an unusual test for this purpose. Again, it is a test not an appropriate for patients treated with LMWHs or factor Xa inhibitors. Thrombin generation assay and F1+2 peptide dosage would have been a better option for this purpose.

4 Page 4, line 147. Wilcoxon paired comparison is better to use here because the data are not normally distributed as it appears in the Table 1.

5 Page 4, line 154. Why were these patients (cancer stroke) treated with anticogulants? The authors should better explain this point. Otherwise, this therapeutic decision appears to be rather empirical. Was the reason related to the aim of the study? Cancer stroke is not the same condition as that of AF stroke, i.e. it not recognize blood stasis, and rather a hypercoagulative/inflammatory state is the main pathophysiological background. I am not sure at all that this is a good ethical approach.

6 Page 8, line 229. It is obvious that Xa inhibitors reduce factor Xa activity! This is the reason why factor Xa test is not appropriate for this study.

7 Page 9, line 233. The Authors should consider the inflammatory state, which is typical, in general, of a neoplastic condition. Anti-Xa inhibitors, unlike LMWH, do not have the properties of reducing inflammation. A comment is required here.

8 Page 9, line 250. I have some doubts on the utility of these tests in the single patient. Certainly, these tests are not appropriate for the daily clinical activity.

  1. Page 9, line 261. Yes, this is a speculation and should be avoided.
  2. Page 9, line 263. The Authors should consider that patients with cancer were treated with chemotherapy, which may have been the cause of the abnormalities of both the coagulative tests and the platelet count.
  3. Page 9, line 275. D-Dimer is not a test to be used for changing an anticoagulant, in general, since it is a sensitive test but with poor specificity. Therefore, D-Dimer, especially in cancer patients, is a mirror of the inflammatory state of these patients.

Author Response

Response to Reviewers’ comments

December 18, 2021

Manuscript ID: jcm-1497215

Title: Role of Factor Xa-Independent Pathway and Anticoagulant therapies in Cancer-related Stroke

Dear reviewers & editor

First of all, we would like to thank the reviewers and editorial board members for their time and valuable comments on our manuscript. We have addressed our opinions on each comment from two reviewers in this response letter, and made several changes and corrections to our original manuscript.

We tried to revise our manuscript according to reviewer’s suggestions as much as possible, and revised parts are written in red color texts in the manuscript. We hope that the revisions in the manuscript and our accompanying responses will be sufficient to make our manuscript suitable for publication in Journal of Clinical Medicine.

We shall look forward to hearing from you at your earliest convenience.

Sincerely yours,

Oh Young Bang, MD, PhD

Department of Neurology, Samsung Medical Center, Sungkyunkwan University,

50 Irwon-dong, Gangnam-gu, Seoul 135-710, Korea

Tel.: 82-2-3410-3599; Fax: 82-2-3410-1430;

Reviewer 3

I have read with interest this study dealing with both cancer and non-cancer stroke. Although the sample size is rather small, results deserve attention. However, I have some critical points to be addressed to the Authors.

Comment 1. Page 2, line 55. Direct Oral Anticoagulants (DOAC) is a better term than NOAC.

Answer 1. Thank you for this helpful suggestion. As per your suggestion, all the terms of NOAC in the manuscript have been changed to DOAC.

Comment 2. Page 2, line 60. The authors should be remember here that some DOAC (Edoxaban and Rivaroxaban) confer an increased risk of bleeding in patients with cancer of the GI and Urinary tracts. References related to those studies should be added.

Answer 2. Thank you for such insightful advice. Following the reviewer’s suggestion, we revised our manuscript from,

“Recent randomized controlled trials comparing the effects of LMWHs and DOACs showed that DOACs could be an alternative option for the treatment of cancer patients with VTE.”

To

“Although rivaroxaban and edoxaban showed an increased bleeding risk especially in gastrointestinal cancer [2], recent randomized controlled trials comparing the effects of LMWHs and DOACs showed that DOACs could be an alternative option for the treatment of cancer patients with VTE.”

At page # 2, paragraph # 2 in Introduction section.

Comment 3. Page 4, line 125. Why did the authors choose factor Xa activity as a test for a hypercoagulative condition? It is an unusual test for this purpose. Again, it is a test not an appropriate for patients treated with LMWHs or factor Xa inhibitors. Thrombin generation assay and F1+2 peptide dosage would have been a better option for this purpose.

Answer 3. We fully agree with the reviewer’s concern. In this study, the D-dimer, which is most widely used to evaluate hypercoagulative condition in cancer stroke, was measured for hypercoagulability whereas prothrombin fragment 1.2 (F1.2) was not measured. We focused on factor Xa activity, a target of DOAC, and NETosis, which is recently known to be related to cancer coagulability. Therefore, we added the following sentence in the limitation section.

“In this study, the level of prothrombin fragment 1.2 (F1.2) was not measured. It deserves further studies on the changes of prothrombin fragment 1.2 (F1.2) with the use of DOACs and LMWH in cancer stroke.”

At page # 8, in Limitation section.

Comment 4. Page 4, line 147. Wilcoxon paired comparison is better to use here because the data are not normally distributed as it appears in the Table 1.

Answer 4. Thank you for this helpful suggestion. As per your suggestion, we re-analyzed Wilcoxon paired comparison and revised our manuscript, p-value of Figure 2, and Supplemental Data 1.

From

“A paired t-test was used to compare changes in hypercoagulability measurements before and after anticoagulation treatment.”

To

“A paired t-test and Wilcoxon signed rank test was used to compare changes in hypercoagulability measurements before and after anticoagulation treatment.”

At page # 4, paragraph # 7 in Methods section.

Data 1. Laboratory findings associated with recurrence of cancer stroke and treatment-related changes.

Situation 1

Situation 2

P

Recurrence of cancer stroke

Stable

Recurrence

D-dimer, μg/mL (n=10)

8.80±7.31

28.11±21.22

0.002

Factor Xa, % (n=9)

69.11±9.96

62.33±18.32

0.164

cfDNA, ng/ml (n=6)

51.56±9.94

65.70±13.49

0.022

Cancer stroke

Enoxaparin

Pre-treat

Post-treat

D-dimer, μg/mL (n=32)

16.05±13.28

8.58±12.26

0.011

Factor Xa, % (n=28)

71.35±25.15

71.92±19.56

0.858

cfDNA, ng/ml (n=15)

53.69±18.86

52.98±11.41

0.495

Cancer stroke

Edoxaban

Pre-treat

Post-treat

D-dimer, μg/mL (n=12)

16.68±10.48

13.08±9.43

0.071

Factor Xa, % (n=12)

77.44±15.49

68.94±14.61

0.019

cfDNA, ng/ml (n=9)

56.73±6.54

71.38±12.30

<0.001

AF stroke

Factor Xa inhibitor

Pre-treat

Post-treat

D-dimer, μg/mL (n=16)

1.53±1.76

0.37±0.16

0.009

Factor Xa, % (n=35)

81.92±15.10

75.71±14.21

0.002

cfDNA, ng/ml (n=15)

41.97±7.75

38.05±4.04

0.096

Comment 5. Page 4, line 154. Why were these patients (cancer stroke) treated with anticogulants? The authors should better explain this point. Otherwise, this therapeutic decision appears to be rather empirical. Was the reason related to the aim of the study? Cancer stroke is not the same condition as that of AF stroke, i.e. it not recognize blood stasis, and rather a hypercoagulative/inflammatory state is the main pathophysiological background. I am not sure at all that this is a good ethical approach.

Answer 5. As mentioned by the reviewers, we agree that cancer stroke and AF stroke are not the same condition. However, in the case of ESUS, hypercoagulability is an important stroke mechanism [3-5], and in this study, all cases with conventional stroke mechanisms were excluded. Therefore, anticoagulants were used as recommended in cancer associated coagulopathy rather than antiplatelet agents. And we have already mentioned in the Limitation section about not investigating cancer stroke patients with antiplatelet in our study.

“Fifth, in this study the role of antiplatelets in factor Xa dependent and independent pathways was not investigated. Many centers treat patients with cancer-related stroke with antiplatelet therapy. In addition, a recent subgroup analysis of the NAVIGATE-ESUS trial found that anticoagulation with rivaroxaban was not superior and perhaps even inferior to antiplatelet therapy with aspirin in patients with history of cancer and ESUS. Further studies are needed.”

And we revised our manuscript to reduce this confusion for our readers.

From

“In patients with cancer stroke, anticoagulation treatment was considered as the first-line antithrombotic treatment in the absence of obvious contraindications.”

To

“In patients with cancer stroke, patients with conventional mechanisms were excluded and hypercoagulable state was confirmed. Therefore, anticoagulation treatment was considered as the first-line antithrombotic treatment in the absence of obvious contraindications.”

At page # 4, paragraph # 6 in Methods section.

Comment 6. Page 8, line 229. It is obvious that Xa inhibitors reduce factor Xa activity! This is the reason why factor Xa test is not appropriate for this study.

Answer 6. Again, we fully agree with the reviewer’s concern. We measured factor Xa activity in cancer stroke patients to test whether factor Xa inhibitors effectively reduce both D-dimer and factor Xa levels in the same way in both cancer stroke and AF-stroke patients. As aforementioned, further studies are needed measuring the level of prothrombin fragment 1.2 (F1.2) in cancer stroke.

Comment 7. Page 9, line 233. The Authors should consider the inflammatory state, which is typical, in general, of a neoplastic condition. Anti-Xa inhibitors, unlike LMWH, do not have the properties of reducing inflammation. A comment is required here.

Answer 7. Thank you for such insightful advice. As per your suggestion, we added following sentence in Discussion section.

“In addition, in previous studies, LMWH was shown to have anti-inflammatory activity unlike factor Xa inhibitors [6].

At page # 7, paragraph # 3 in Discussion section.

Comment 8. Page 9, line 250. I have some doubts on the utility of these tests in the single patient. Certainly, these tests are not appropriate for the daily clinical activity.

Answer 8. We fully agree with the reviewer’s concern. Among these tests, factor Xa activity and cfDNA level are difficult to be implemented as a daily clinical activity and should be performed for research purposes. Recently, several studies are underway for point-of-care tests to measure or predict coagulopathy in cancer patients.

Comment 9. Page 9, line 261. Yes, this is a speculation and should be avoided.

Answer 9. We fully agree with the reviewer’s opinion on this. we revised our manuscript from,

“We speculated that factor Xa activity was decreased after stroke as a result of an increased consumption of coagulation factors, similar to disseminated intravascular coagulation. A longer PT/INR and aPTT and a lower platelet count in cancer stroke compared with cancer controls support this explanation.”

To

“Therefore, it is difficult to accurately explain this finding, but it may be related to a longer PT/INR and aPTT and a lower platelet count in cancer stroke compared with cancer controls.”

At page # 8, paragraph # 4 in Discussion section.

Comment 10. Page 9, line 263. The Authors should consider that patients with cancer were treated with chemotherapy which may have been the cause of the abnormalities of both the coagulative tests and the platelet count.

Answer 10. I totally agree with your opinion. The detailed analysis about the type of chemotherapy used deserve further studies because chemotherapeutic agents can cause endothelial dysfunction resulting atherosclerotic/lacunar stroke as well as coagulopathy. As aforementioned, this has been described in the Limitation section as follows.

“Chemotherapy induces cell death and DNA release into the plasma, but the duration, type, and regimen of chemotherapy was not considered in our analysis.”

At page # 8, in Limitation section.

Round 2

Reviewer 1 Report

The authors have responded satisfactorily to my comments.

Reviewer 3 Report

The authors' answers are in line with my requests in this revised form, I have no other points of concern to be addressed to them.